# ASSOCIATIVE TRANSFORMER IS A SPARSE REPRESENTATION LEARNER

## ABSTRACT

Emerging from the monolithic pairwise attention mechanism in conventional Transformer models, there is a growing interest in leveraging sparse interactions that align more closely with biological principles. Approaches including the Set Transformer and the Perceiver employ cross-attention consolidated with a latent space that forms an attention bottleneck with limited capacity. Building upon recent neuroscience studies of Global Workspace Theory and associative memory, we propose the **A**ssoc**i**ative **T**ransformer (AiT). AiT induces low-rank explicit memory that serves as both priors to guide bottleneck attention in the shared workspace and attractors within associative memory of a Hopfield network. Through joint end-to-end training, these priors naturally develop module specialization, each contributing a distinct inductive bias to form attention bottlenecks. A bottleneck can foster competition among inputs for writing information into the memory. We show that AiT is a sparse representation learner, learning distinct priors through the bottlenecks that are complexity-invariant to input quantities and dimensions. AiT demonstrates its superiority over methods such as the Set Transformer, Vision Transformer, and Coordination in various vision tasks.

## 1 INTRODUCTION

The predominant paradigm in conventional deep neural networks has been characterized by a monolithic architecture, wherein each input sample is subjected to uniform processing within a singular model framework. For instance, Transformer models use pairwise attention to establish correlations among disparate segments of input information Vaswani et al. (2017); Dosovitskiy et al. (2021). Emerging from the pair-wise attention mechanism, there is a growing interest in leveraging modular and sparse interactions that align more closely with biological principles. This sparsity attribute has demonstrated advantages in enhancing model performance and learning efficiency, making it a crucial element for intelligent entity learning Brooks (1991); Greff et al. (2020); Minsky (1986).

Modularization of knowledge can find resonance with the neuroscientific grounding of the Global Workspace Theory (GWT) Baars (1988); Dehaene S. (1998); VanRullen & Kanai (2020); Juliani et al. (2022). GWT explains a fundamental cognitive architecture for information processing within the brain, where diverse specialized modules compete to write information into a shared workspace through a communication bottleneck. The bottleneck facilitates the processing of content-addressable information through attention that is guided by working memory Awh et al. (2006); Gazzaley & Nobre (2011). The coordination method Goyal et al. (2022b) represents the initial attempt to assess the effectiveness of GWT in conventional neural network models. Unfortunately, this method relies on iterative cross-attention for both information writing and retrieval within the shared workspace. When examining information retrieval in the human brain, it is evident that memory typically encompasses both working memory and long-term memory in the hippocampus. Specifically, the hippocampus operates on Hebbian learning for retrieving information from working memory, akin to the associative memory found in Hopfield networks Hopfield (2007); Ramsauer et al. (2021). Our research has revealed that replacing such a repetitive attention-based mechanism with a consolidated, more biologically-plausible associative memory can lead to improved model performance. Associative memory has the capability to directly store and retrieve patterns from the shared workspace without the need for additional parameters by relying on an energy function, which fundamentally differs from an attention mechanism. Our objective is to introduce a shared

workspace augmented with associative memory into a Transformer model, thereby facilitating a more comprehensive and efficient association of information fragments.

To this end, we propose the **A**ssociative **T**ransformer (AiT) based on a novel global workspace layer augmented by associative memory. The global workspace layer entails three main components: 1) the squash layer: input data is transformed into a list of patches regardless of which samples they come from, 2) the bottleneck attention: patches are sparsely selected to learn a set of priors in low-rank memory based on a bottleneck attention mechanism, and 3) the Hopfield network: information is broadcast from the shared workspace to update the current input based on the associative memory of a Hopfield network. Moreover, the bottleneck attention and the low-rank memory contributes to reduced model complexity. However, cascading multiple of these components may lead to difficulty in the emergence of specialized priors in explicit memory. As information flows through multiple layers, it becomes more challenging to maintain specialized priors from diluted representations. Consequently, learning specialized priors in layers cascaded in depth requires a mechanism that counteracts this inherent loss of input specificity. To overcome this challenge, we propose the bottleneck attention balance loss to encourage the diverse selection of inputs in the shared workspace. Through end-to-end training, we show the emerging specialization of low-rank priors, contributing to enhanced performance in vision tasks. This distinguishes our work from previous literature, which relied on latent memory comprising indistinct priors with the same dimension as the input, such as Set Transformer Lee et al. (2019), Perceiver Jaegle et al. (2021), and Luna Ma et al. (2021). The no-free-lunch theorem Baxter (2000); Goyal & Bengio (2020b) states that a set of inductive bias over the space of all functions is necessary to obtain generalization. We demonstrate that the specialization of priors serves as critical inductive biases, encouraging competition among input data and inducing sparsity in the attention mechanism of Transformer models.

Overall, the main contributions of this work are as follows. (1) This work proposes a more biologically plausible learning framework called Associative Transformer (AiT) based on the Global Workspace Theory and associative memory. (2) AiT is a sparse representation learner, leveraging sparse bottleneck attention enhanced by a novel attention balance loss to acquire naturally emerging specialized priors. (3) We devise low-rank priors that are adaptively encoded and decoded for increased memory capacity. AiT can learn a large set of specialized priors (up to 128) from a diverse pool of patches (up to 32.8k). (4) The learned priors serve as attractors within the associative memory of a Hopfield network, enabling information broadcast from the workspace. This is the first work to incorporate the Hopfield network as an integral element in a sparse attention mechanism.

## 2 RELATED WORK

This section provides a summary of relevant research concerning sparse attention architectures. We investigate and compare these studies based on their relatedness to the global workspace theory in terms of several key conditions (please see Appendix A.2 for a complete comparison).

Transformer models do not possess inductive biases that allow the model to attend to different segments of the input data Goyal & Bengio (2020a). To enhance Transformer models, studies of sparse attention architectures explored consolidating latent memory to extract contextual representations from input data Gupta & Berant (2020); Jaegle et al. (2021); Goyal et al. (2022b); Jaegle et al. (2022); Lee et al. (2019); Ma et al. (2021). For instance, Perceiver Jaegle et al. (2021) and Perceiver IO Jaegle et al. (2022) used iterative cross-attention with a latent array as priors and a latent transformation applied to the priors, to capture dependencies across input data. Set Transformer Lee et al. (2019) and Linear Unified Nested Attention (Luna) Ma et al. (2021) employed iterative cross-attention, but without using a latent transformation. Other attention mechanisms that rely on strong inductive biases with predefined network modularization are omitted Qiu et al. (2020). In our method, distinct priors naturally emerge through end-to-end training. Moreover, the previous methods using latent memory necessitated priors with the same dimension as the input. In contrast, we devise low-rank priors that can be encoded and decoded adaptively for increased memory capacity.

In the same vein of building sparse attention mechanisms through a shared workspace, Coordination Goyal et al. (2022b) used iterative cross-attentions via a bottleneck to encourage more effective module communication. They argued that more flexibility and generalization could emerge through the competition of specialized modules. However, the priors in the coordination method possess the same dimension as the input, and the number of priors is limited to fewer than 10. The evaluation was also restricted to simple tasks. Unlike the coordination method, we propose low-rank explicit

memory to learn a larger set of specialized priors (up to 128) from a pool of patches (up to 32.8k). Moreover, the coordination method relies on iterative cross-attentions to learn such priors, while this work focuses on a novel learning method of associative memory-augmented attention.

Furthermore, external memory such as tape storage and associative memory has been successfully employed Graves et al. (2014); Gülçehre et al. (2018); Krotov & Hopfield (2016); Hoover et al. (2023). Recent studies explored the potential use of Hopfield networks Hopfield (2007) and their modern variants Demircigil et al. (2017); Ramsauer et al. (2021) in Transformers. In contrast to these investigations, we incorporate Hopfield networks as an integral element in constructing the global workspace layer, functioning as a mechanism for information broadcast in the shared workspace. This goal is fundamentally different from prior studies focused on using Hopfield networks independently of the attention mechanism.

## 3 INSPECTING ATTENTION HEADS IN VISION TRANSFORMERS

Vision Transformers (ViT) tackle image classification tasks by processing sequences of image patches. The pre-processing layer partitions an image into non-overlapping patches, followed by a learnable linear projection layer. Let $x \in \mathbb{R}^{H \times W \times C}$ be an input, where $(H, W)$ is the resolution of the image and $C$ is the number of channels. $x$ is separated into a sequence of patches $x_p \in \mathbb{R}^{N \times (P^2 \cdot C)}$, where $(P, P)$ is the resolution of each image patch and $N = \frac{HW}{P^2}$ is the number of patches. These patches are mapped to embeddings $v_p \in \mathbb{R}^{N \times E}$ with the linear projection. ViT leverages self-attention where each head maps a query and a set of key-value pairs to an output. The patch embeddings are used to obtain the query, key, and value based on linear transformations $W^Q \in \mathbb{R}^{E \times D}$, $W^K \in \mathbb{R}^{E \times D}$, and, $W^V \in \mathbb{R}^{E \times D}$. The output is a weighted sum of the values:

$$\mathrm{h}^i(v) = \mathrm{softmax}(\frac{W_i^Q v (W_i^K v)^T}{\sqrt{D}}) \, W_i^V v, \tag{1}$$

$$\mathrm{Multi\text{-}head}(v) = \mathrm{Concat}(\mathrm{h}^1, \ldots, \mathrm{h}^A) \, W^O, \tag{2}$$

where $W^O$ is a linear transformation for outputs, and $A$ is the number of attention heads.

We assume that the competition within the pair-wise attention of different patches would be of importance for the model to learn meaningful representations. If such competition exists, a trained model will naturally result in sparser interactions in attention heads. Therefore, we first performed an analysis of the operating modes of different attention heads in a pretrained ViT model by measuring the number of patches each head is attending to. We refer to Appendix A.4 for the detailed experimental settings. The inspection revealed the existing competition among patches and a large redundancy in the pair-wise attention. Less than 80% interactions were activated in ViT, and several heads from the middle layers used only 50% or less interactions with higher sparsity compared to the other layers. Based on the observation, by introducing a bottleneck that limits each attention head's focus to foster competition, we obtain inductive biases for more efficient patch learning.

## 4 ASSOCIATIVE TRANSFORMER

This section discusses the essential building blocks of the Associative Transformer (AiT), where patches compete to write into the shared workspace through bottleneck attention. The workspace enables an efficient information writing and reading mechanism by learning a set of priors in explicit memory. These priors are low-rank and learned progressively from the input through end-to-end training. The priors guide the bottleneck attention with an emerging specialization property. Moreover, we extend the learned priors to attractors within the associative memory of a Hopfield network, facilitating information retrieval from memory and efficient association of information fragments.

### 4.1 GLOBAL WORKSPACE LAYER

We devise an associative memory-augmented attention layer called the *global workspace layer*, which comprises the squash layer, the bottleneck attention guided by low-rank memory, and the information retrieval within the associative memory of a Hopfield network (Figure 1). The global workspace layer can be seen as an add-on component on the monolithic Vision Transformer, where the feed-forward layers process patches before they enter the workspace, facilitating abstract relation learning, and the self-attention learns the contextual relations for a specific sample. The global workspace layer learns spatial relations across various samples and time steps.

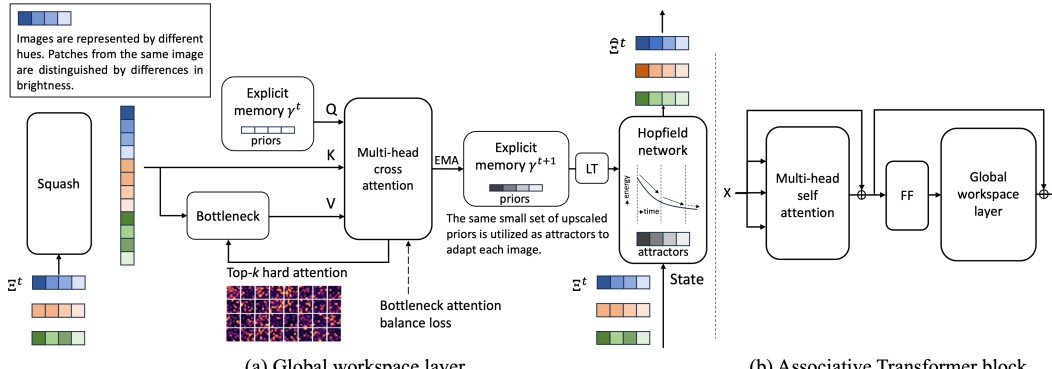

(a) Global workspace layer    (b) Associative Transformer block

Figure 1: The scheme of the Associative Transformer. (a) In a global workspace layer, the input $\mathbb{R}^{B \times N \times E}$ is squashed into vectors $\mathbb{R}^{(B \times N) \times E}$. The squashed representations are projected to a low-rank latent space of dimension $D << E$ and then are sparsely selected and stored in the explicit memory via a fixed bottleneck $k << (B \times N)$. The Hopfield network utilizes the memory to reconstruct the input, where a learnable linear transformation (LT) scales the memory contents to match the input dimension $E$. (b) The Associative Transformer block consists of sequentially connected self attention, feed-forward layers, and the global workspace layer.

**Squash layer**    In self-attention, patches from the same sample are attended to. In our work, we improve the diversity in patch-wise correlation learning beyond one sample using a *squash layer*. The squash layer obtains patch representations from the entire training batch to enable competition among patches not only from the same sample but also from different samples. This differs from traditional approaches where the competition resides within specific samples. The squash layer concatenates patches within one batch $V \in \mathbb{R}^{B \times N \times E}$ into vectors $V \in \mathbb{R}^{(B \times N) \times E}$, which forms a list of patches regardless of the samples they are from. Though the number of patches changes in practice depending on the batch size, the communication bottleneck with a fixed capacity $k$ limits the number of patches the workspace can attend to at any given time. Since the bottleneck decreases the complexity from $O((B \times N)^2)$ to $O((B \times N) \times k)$, using the squash layer increases the diversity of input patches without adding to the complexity. With the greater diversity, a sample's classification task, for instance, can benefit from other patches belonging to the same class within the batch input.

**Low-rank explicit memory**    An explicit memory bank with limited slots aims to learn $M$ *priors* $\gamma = \mathbb{R}^{M \times D}$ where $D$ is the dimension of the prior. The priors in the memory bank are used as various keys to compute the bottleneck attentions that extract different sets of patches from the squashed input. Furthermore, using low-rank priors reduces memory consumption, as a lower dimension $D << E$ is obtained through a down-scale linear transformation.

## 4.2    BOTTLENECK ATTENTION WITH A LIMITED CAPACITY

The objective of the bottleneck attention is to learn a set of priors that guide attention to various input patches. This is enabled by a cross-attention mechanism constrained by hard attention. We first consider a tailored cross-attention mechanism to update the memory bank based on the squashed input $\Xi^t = V^t \in \mathbb{R}^{(B \times N) \times E}$, then we discuss the case of limiting the capacity via a top-$k$ hard attention. Notably, in the cross-attention, the query is a function of the current memory content $\gamma^t = \{\gamma_i^t\}_{i=1}^M$. The key and value are functions of the squashed input $\Xi^t$. The attention scores for head $i$ can be computed by $A_i^t(\gamma^t, \Xi^t) = \mathrm{softmax}(\frac{\gamma^t W_{i,t}^Q (\Xi^t W_{i,t}^K)^T}{\sqrt{D}})$. This is the case of soft attention with limited constraints on the bottleneck capacity. Moreover, the hard attention allows patches to compete to enter the workspace through a $k$-size bottleneck, fostering the selection of essential patches. In particular, the top-$k$ patches with the highest attention scores from $A_i^t$ are selected to update the memory. To ensure a stable update across different time steps, we employ the layer normalization and the Exponentially Weighted Moving Average (EWMA) method as follows

$$\mathrm{head}_i^t = \text{top-}k(A_i^t)\Xi^t W_t^V, \ \hat{\gamma}^t = \mathrm{LN}(\mathrm{Concat}(\mathrm{head}_1^t, \ldots, \mathrm{head}_A^t)W^O), \qquad (3)$$

$$\gamma^{t+1} = \alpha \cdot \gamma^t + (1 - \alpha) \cdot \hat{\gamma}^t, \ \gamma^{t+1} = \frac{\gamma^{t+1}}{\sqrt{\sum_{j=1}^M (\gamma_j^{t+1})^2}}, \qquad (4)$$

where top-$k$ selects the $k$ highest attention scores, LN is the layer normalization, and $\alpha$ is a smoothing factor determining the decay rate of older observations. EWMA ensures the stable memory update with varying batch sizes by accumulating both old $\gamma^t$ and new memories $\hat{\gamma}^t$.

During the test time, the explicit memory is frozen, functioning as fixed priors, and any memory update from the bottleneck attention will not be retained (Figure 8). We only compute $\gamma^{t+1}$ for the following pattern retrieval step in Hopfield networks for the current batch. To ensure a fair evaluation on the test dataset, the same explicit memory from the training time is utilized across all test batches.

**Bottleneck attention balance loss**   The bottleneck attention and the low-rank memory contribute to reduced model complexity of the global workspace layer. Nevertheless, employing multiple components cascaded in depth might lead to difficulty in the emergence of specialized priors in the explicit memory (Figure 9). To overcome this challenge, we propose the bottleneck attention balance loss to encourage the selection of diverse patches from different input positions. The bottleneck attention balance loss $\ell_{\text{bottleneck}}$ comprises two components, i.e., the accumulative attention scores and the chosen instances for each input position. Then, we derive the normalized variances of the two metrics across different positions as follows

$$\ell_{\text{loads}_{i,l}} = \sum_{j=1}^{M} (\mathrm{A}_{i,j,l}^t > 0), \;\; \ell_{\text{importance}_{i,l}} = \sum_{j=1}^{M} \mathrm{A}_{i,j,l}^t, \tag{5}$$

$$\ell_{\text{bottleneck}_i} = \frac{\text{Var}(\{\ell_{\text{importance}_{i,l}}\}_{l=1}^{B \times N})}{(\frac{1}{B \times N} \sum_{l=1}^{B \times N} \ell_{\text{importance}_{i,l}})^2 + \epsilon} + \frac{\text{Var}(\{\ell_{\text{loads}_{i,l}}\}_{l=1}^{B \times N})}{(\frac{1}{B \times N} \sum_{l=1}^{B \times N} \ell_{\text{loads}_{i,l}})^2 + \epsilon}, \tag{6}$$

where $\mathrm{A}_{i,j,l}^t$ denotes the attention score of the input position $l$ for the $j$th memory slot of head $i$, $\ell_{\text{importance}}$ represents the accumulative attention scores for all $M$ memory slots concerning each input position, $\ell_{\text{loads}}$ represents the chosen instances for each input position in $M$ memory slots, $\text{Var}(\cdot)$ denotes the variance, and $\epsilon$ is a small value to avoid division by zero. Finally, the loss scores for all the heads are summed up as follows: $\ell_{\text{bottleneck}} = \sigma \cdot \sum_{i=1}^{A} \ell_{\text{bottleneck}_i}$ where $\sigma$ is a coefficient.

## 4.3   Information retrieval within associative memory

After writing information into the shared workspace, the learned priors can serve as attractors within associative memory. The objective is to reconstruct the current input patches towards more globally meaningful representations based on these attractors.

**Attractors**   Priors learned in the memory bank act as attractors in associative memory. Attractors have basins of attraction defined by an energy function. Any input state that enters an attractor's basin of attraction will converge to that attractor. The attractors in associative memory usually have the same dimension as input states; however, the priors $\gamma^{t+1}$ in the memory bank have a lower rank compared to the input. Therefore, we employ a learnable linear transformation $f_{\text{LT}}(\cdot)$ to project the priors into a space of the same dimension, $E$, as the input before using them as attractors.

**Retrieval using the energy function in Hopfield networks**   Hopfield networks have demonstrated their potential as a promising approach to constructing associative memory. In particular, a continuous Hopfield network Demircigil et al. (2017); Ramsauer et al. (2021) operates with continuous input and output values. The upscaled priors $f_{\text{LT}}(\gamma^{t+1})$ are stored within the continuous Hopfield network and are subsequently retrieved to reconstruct the input state $\Xi^t$. Depending on an inverse temperature variable $\beta$, the reconstructed input $\hat{\Xi}^t$ can be either a metastable state that represents a mixture of various attractors or a fixed state represented by one of the attractors. A large $\beta$ makes it less likely for metastable states to appear, while a small $\beta$ increases the likelihood. The continuous Hopfield network employs an energy function to enable the evolution of patches into more globally meaningful representations with respect to the learned attractors. We update each patch representation $\xi^t \in \Xi^t$ by decreasing its energy $E(\xi^t)$ within associative memory as follows

$$E(\xi^t) = -\text{lse}(\beta, f_{\text{LT}}(\gamma^{t+1})\xi^t) + \frac{1}{2}\xi^t \xi^{tT} + \beta^{-1}\log M + \frac{1}{2}\zeta^2, \tag{7}$$

$$\zeta = \max_i |f_{\text{LT}}(\gamma_i^{t+1})|, \;\; \hat{\xi}^t = \arg\min_{\xi^t} E(\xi^t), \tag{8}$$

where lse is the log-sum-exp function and $\zeta$ denotes the largest norm of attractors. Equation 7 describes an iteration that can be applied several times. Usually, we apply just a single step for

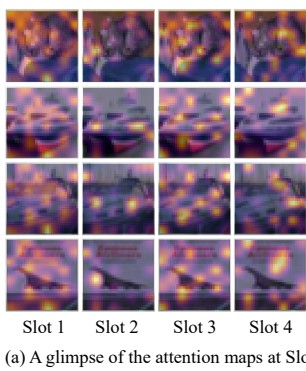

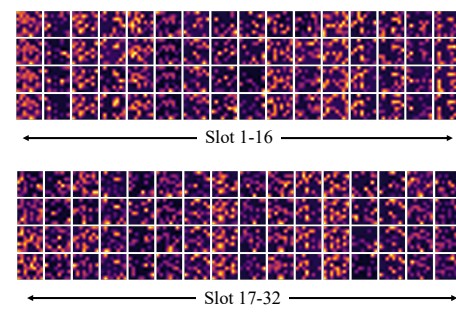

Slot 1-16

Slot 17-32

(a) A glimpse of the attention maps at Slot 1 to Slot 4 of four distinct input images.

(b) The attention maps for all 32 slots in the memory bank, applied to four distinct input images. Each memory slot learned to attend to different regions of pixels in input images.

Slot 1     Slot 2     Slot 3     Slot 4

Figure 2: Learned distinct memory slot attentions in AiT. Each slot's activation maps highlight a specific area during the selection of relevant image patches.

efficient forward and backward computation during end-to-end training. $t$ is the batch time step, and the iteration time step is implicit. Additionally, a skip connection functioning as the information broadcast from the global workspace is employed to obtain the final output $\Xi^{t+1} = \hat{\Xi}^t + \Xi^t$.

## 5    EXPERIMENTS

In this section, we discuss the settings and extensive empirical results for image classification and relational reasoning tasks. Our study demonstrates that AiT outperforms the coordination method and other sparse attention-based approaches in terms of both performance and model complexity.

### 5.1    SETUP

**Datasets**    We evaluate model performance on two different scales of datasets (1) small (Triangle Goyal et al. (2022b), CIFAR10 Krizhevsky (2009), and CIFAR100 Krizhevsky & Hinton (2009)) and (2) middle (Oxford-IIIT Pet Parkhi et al. (2012) and Sort-of-CLEVR Santoro et al. (2017)). We train the model on these datasets from scratch using the training split and evaluate using the test split. A detailed description of the datasets can be found in Appendix A.1.

**Model variants**    We investigate three different sizes of model configurations, i.e., Small, Medium, and Base. The Base variant setting is adapted from Vision Transformer (ViT) using 12 layers, 12 attention heads for each layer, a hidden dimension of 768, and an MLP dimension of 3072. The Medium variant with 6 layers and the Small variant with 2 layers are added for efficiency comparisons among approaches. The CLS token is removed while the pooled representations of the last dense network layer are used instead since using the CLS token leads to undermined learning results in vision tasks Wang et al. (2021); Graham et al. (2021).

**Hyperparameters**    The hyperparameters were chosen based on a grid search. A batch size of 512 was employed for the CIFAR datasets and the Triangle dataset, 128 for the Pet dataset, and 64 for the Sort-of-CLEVR dataset. We utilized the AdamW optimizer with $\beta_1 = 0.9$, $\beta_2 = 0.999$, and a weight decay of 0.01. A cosine learning rate scheduler was implemented with an initial learning rate of 1e-5, a warm-up phase of 5 (15) epochs within a total of 100 (300) epochs, and a minimum learning rate set to 1e-6. The smoothing factor of the exponentially weighted moving average, the coefficient $\sigma$, and the small value $\epsilon$ in the bottleneck balance loss were set to 0.9, 1e-2, and 1e-10, respectively. For AiT, we employed a memory slot size of 32 and a bottleneck attention head size of 8. We used a bottleneck size of 512 for CIFAR and Pet, 64 for Triangle, and 256 for Relational Reasoning. We used 32 memory slots for CIFAR, Triangle, and Relational Reasoning, and 128 slots for Pet (Appendix A.3). Unless otherwise noted, we trained the model for 100 epochs and reported the mean of three individual experiments. The code will be made publicly available.

### 5.2    CLASSIFICATION TASKS

The experiments on image classification tasks include comparisons to a wide range of methods (Table 1). We used the author-recommended hyperparameters to re-implement these methods. Regarding the coordination method, we have examined the efficacy of its variants with different model configurations. The default coordination model consists of 4 layers, with parameter sharing among

Table 1: Performance comparison in image classification tasks

| Methods | CIFAR10 | CIFAR100 | Triangle | Average | Model Size |
|---|---|---|---|---|---|
| AiT-Base | **85.44** | **60.78** | 99.59 | **81.94** | 91.0 |
| AiT-Medium | 84.59 | 60.58 | 99.57 | 81.58 | 45.9 |
| AiT-Small | 83.34 | 56.30 | 99.47 | 79.70 | 15.8 |
| Coordination Goyal et al. (2022b) | 75.31 | 43.90 | 91.66 | 70.29 | 2.2 |
| Coordination-DH | 72.49 | 51.70 | 81.78 | 68.66 | 16.6 |
| Coordination-D | 74.50 | 40.69 | 86.28 | 67.16 | 2.2 |
| Coordination-H | 78.51 | 48.59 | 72.53 | 66.54 | 8.4 |
| ViT-Base Dosovitskiy et al. (2021) | 83.82 | 57.92 | **99.63** | 80.46 | 85.7 |
| ViT-Small | 79.53 | 53.19 | 99.47 | 77.40 | 14.9 |
| Perceiver Jaegle et al. (2021) | 82.52 | 52.64 | 96.78 | 77.31 | 44.9 |
| Set Transformer Lee et al. (2019) | 73.42 | 40.19 | 60.31 | 57.97 | 2.2 |
| BRIMs Mittal et al. (2020) | 60.10 | 31.75 | - | 45.93 | 4.4 |
| Luna Ma et al. (2021) | 47.86 | 23.38 | - | 35.62 | 77.6 |

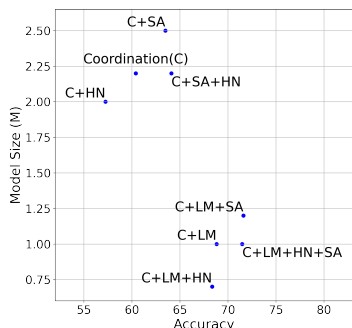

Figure 4: Model size vs. accuracy for configurations.

different attention layers. Coordination-D is a deeper model with 8 layers using the parameter sharing. Coordination-H is a high-capacity model with 4 layers that employ individual parameters. Coordination-DH is a high-capacity model with 8 layers. The results show that AiT achieved better performance compared to the coordination methods. The AiT performance also increased when scaling it from AiT-Small to AiT-Base, while the coordination methods appeared difficult to scale with the increasing number of layers and parameters, as seen in the case of Coordination-DH. Moreover, AiT outperformed the other baseline methods, demonstrating strong performance. For instance, compared to ViT-Base with 85.7M parameters, AiT-Medium is a shallower model with only 45.9M parameters. Nevertheless, AiT-Medium exhibited an average performance of 81.58%, surpassing the ViT-Base model's average of 80.46% and requiring much fewer parameters. AiT also outperformed sparse attention-based methods such as Perceiver and Set Transformer.

We extended the evaluation to a middle-sized dataset of Oxford Pet. We used a patch size of 16. A larger memory of 128 slots was employed due to the higher resolution and the increased data class complexity. For the Oxford Pet dataset, we trained the model for 300 epochs. Figure 3 reveals that ViT performance can be enhanced by including the global workspace layer. AiT-Medium with fewer parameters also outperforms ViT-Base in the Pet dataset. Though AiT-Medium converges at a later training stage, it is a smaller model with fewer layers to compute compared to ViT-Base.

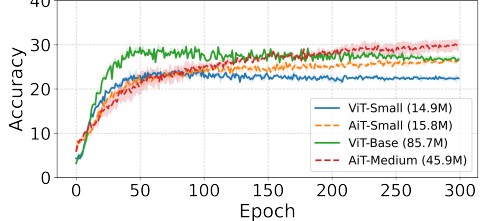

Figure 3: Comparison on the Pet dataset, which shows enhanced accuracy for AiT.

**Prior specialization** Patches in one image can be attended sparsely by different priors. As shown in Section 3, a monolithic Transformer model needs to learn such specialization and relations without the inductive bias introduced by the global workspace layer. Notably, these priors learned to focus on independent spatial areas of an image to guide the attention. We visualized the activation maps for the specialized priors used in CIFAR-10 for AiT-Small (Figure 2). Each slot's activation maps highlight specific areas during the selection of relevant patches.

### 5.3 ABLATION STUDY

We conducted a comprehensive ablation study to gain insights into the functionalities of the various components of AiT (Table 2). In AiT with reset memory, we initialized the explicit memory every epoch. The W/O Hopfield ablation replaces the Hopfield network with another multi-head attention (MHA) that shares the same architecture as the self attention in Figure 1.b. The rationale behind this ablation is grounded in the prior studies of Set Transformer and Perceiver models that relied on two MHA components cascaded in depth. For a fair comparison, instead of simply removing the Hopfield network, we replaced it with the MHA. The added MHA takes the input state $\Xi^t$ as the query, and the upscaled priors $f_{\text{LT}}(\gamma^{t+1})$ as the key and value, i.e., $\hat{\Xi}^t = \text{MHA}(\Xi^t, f_{\text{LT}}(\gamma^{t+1}))$.

Moreover, W/O memory evaluates performance when the global workspace layer is removed, the remaining components of which are equivalent to a simple Vision Transformer. W/O bottleneck shows performance using dense attention by removing the top-$k$ bottleneck capacity constraint. W/O SA examines performance when the multi-head self attention component in Figure 1.b is excluded, and W/O FF evaluates performance when the feedforward component is removed. Lastly, the dense

networks consist of repeated feedforward components with the other components removed in each AiT block. The analysis suggests that the complete model with all components can achieve the highest classification accuracy. The bottleneck appeared to play a significant role in improving performance, since its absence led to an evident decrease in accuracy. Making changes to other components such as Hopfield networks and the explicit memory, while not as impactful, still resulted in degraded accuracy. Despite the relatively good performance of dense networks, their performance in relational reasoning tasks is considerably inferior to that of the AiT model (Section 5.8). We demonstrate the without memory forward ablation in Table 7 and Table 8). The results show that AiT performs as well as or better than the without memory forward ablation.

Table 2: Comparison based on an ablation study. The results indicate that combining all the components leads to the highest performance in all the tasks.

| Models | CIFAR10 | CIFAR100 | Triangle | Average |
|---|---|---|---|---|
| AiT | **83.34** | **56.30** | **99.47** | **79.70** |
| Reset memory | 81.94 | 55.96 | 99.46 | 79.12 |
| W/O Hopfield | 81.03 | 54.96 | 99.44 | 78.48 |
| W/O memory (ViT) | 79.53 | 53.19 | 99.47 | 77.40 |
| Dense networks | 77.78 | 53.14 | 99.46 | 76.79 |
| W/O bottleneck | 75.40 | 46.53 | 93.33 | 73.75 |
| W/O SA | 72.72 | 47.75 | 99.46 | 73.31 |
| W/O FF | 69.51 | 40.89 | 97.61 | 69.34 |

Table 3: Different memory initialization approaches. The Gaussian distribution method was found to give the best results.

| Memory initialization methods | Accuracy |
|---|---|
| Gaussian distribution | **83.34** |
| Positional embedding Vaswani et al. (2017) | 78.51 |
| Uniform distribution Graves et al. (2014) | 81.92 |
| Identity distribution Goyal et al. (2022a) | 78.56 |

## 5.4 COMPARISON WITH THE COORDINATION METHOD

We performed a detailed comparison with the coordination method in terms of test accuracy and model size. Figure 4 depicts the results for CIFAR-10 based on models with a single layer. Notably, using the low-rank memory (LM) that has a more diverse set of priors showed benefits in both improving the performance and decreasing the model size. For instance, the baseline coordination (C) method exhibited moderate accuracy of 60.41% with a model size of 2.2M. In contrast, consolidating the low-rank memory and the self-attention (C+LM+SA) exhibited the highest accuracy of 71.62%, while maintaining a relatively compact size of 1.2M. The Hopfield network (HN) maintained the model performance while reducing the model size by replacing the cross-attention with more efficient information retrieval. However, HN was effective only when either the LM or SA component was applied. We assume that retrieval with the Hopfield associative memory relies on a diverse set of priors, which is enabled by the enhanced bottleneck attention using the low-rank memory and the attention balance loss, and the learning through self-attention. By contrast, the previous coordination method had a limited number of priors, e.g. 8, and did not employ self-attention to correlate among input patches. Moreover, integrating all three components (C+LM+HN+SA) resulted in a competitive accuracy of 71.49% with a compact model size of 1.0M.

## 5.5 MEMORY INITIALIZATION

To initialize the explicit memory, we set each slot with values drawn from a specific distribution. We investigated several memory initialization methods (Table 3). The Gaussian distribution generates random values with a mean of 0 and a variance of 1. The sinusoidal positional embedding Vaswani et al. (2017) uses sine and cosine functions to represent positions in a sequence. The uniform distribution Graves et al. (2014) uses an upper bound $\frac{1}{\sqrt{M+D}}$, where $M$ is the memory slot number and $D$ is the slot size. The identity distribution Goyal et al. (2022a) uses ones on the diagonal and zeros elsewhere. We found that the Gaussian distribution resulted in the best performance, possibly by preventing specific priors from dominating the learning process in early training stages.

## 5.6 EFFICACY OF BOTTLENECK ATTENTION BALANCE LOSS

The Bottleneck Attention Balance Loss facilitates selection of diverse input patches for each prior. To quantitatively measure the efficacy, we computed sparsity scores that represent the ratio of distinct patches in all selected patches. In Figure 10, we observe an apparent increase in the patch diversity.

## 5.7 VARYING THE INVERSE TEMPERATURE IN HOPFIELD NETWORKS

We investigated the effect of the inverse temperature on information retrieval based on the Hopfield networks in Figure 5, which shows the reconstructed patches in the CIFAR-10 task for the AiT-

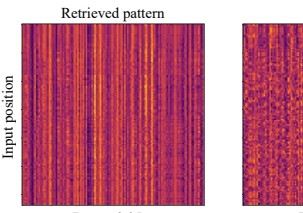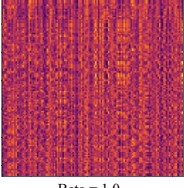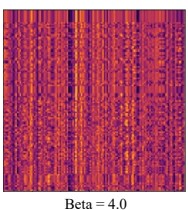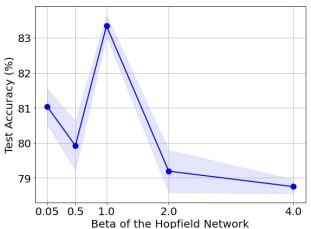

(a) Retrieved patterns from the Hopfield network for the first 128 patch positions in the input.

(b) Test accuracy when applying varying beta scores.

Figure 5: Comparison with varying inverse temperature scores. The inverse temperature beta influences the formation of metastable states that concurrently represent multiple patch representations. A smaller beta is more likely to generate such metastable states, while a larger beta leads to a stronger separation of different patterns. However, a larger beta can also lead to local minima, where input patterns are reconstructed to the same pattern within associative memory.

Small model. We found that using an inverse temperature of 1.0 gave the best retrieval performance based on the Hopfield networks. The results suggest that the beta parameter requires tuning to reach optimal performance. We aim to study a mechanism to adjust the beta adaptively in the future, addressing this sensitivity and potentially further improving performance.

## 5.8 RELATIONAL REASONING

In relational reasoning tasks, we aim to train a model to answer questions concerning the properties and relations of various objects based on a given image. A performant model can attend to specific regions of images for the question-answering task. We employed the Sort-of-CLEVR dataset Santoro et al. (2017) and compared performance to both Transformer based models including Set Transformer and the coordination method, and other non-Transformer based models including CNN+MLP and CNN+Relation Networks (CNN+RN) Santoro et al. (2017). The non-Transformer based models incorporated inductive biases into their architectures, such as convolutional layers focusing on different image areas. This often results in superior performance compared to the Transformer based methods that lack a built-in inductive bias. Moreover, two dense networks, the Dense-Small and Dense-Base, are included as additional non-Transformer based models. The Dense-Small (11.1M) and Dense-Base (62.7M) are derived from the AiT-Small and AiT-Base, respectively. Additionally, in relational reasoning tasks, a question was embedded with an embedding layer that consists of a learnable linear projection and layer normalization before and after the linear projection. The question embedding was then concatenated to image patch embeddings as the input of a model and the labels were a list of answer options with 10 classes.

Table 4 presents the results for relational and non-relational tasks. In the non-relational task, the question pertains to the attributes of a specific object, whereas in the relational task, the question focuses on the relations between different objects. A description of the dataset can be found in Appendix A.1. The results demonstrate a substantial improvement in AiT's performance when addressing the relational reasoning tasks. This indicates that the global workspace layer can

Table 4: Performance comparison on the relational reasoning tasks.

| Methods | Relational | Non-relational | Average |
|---|---|---|---|
| Transformer based models | | | |
| AiT-Small | **76.82** | **99.85** | **88.34** |
| Coordination Goyal et al. (2022b) | 73.43 | 96.31 | 84.87 |
| Set Transformer Lee et al. (2019) | 47.63 | 57.65 | 52.64 |
| Non-Transformer based models | | | |
| CNN+RN Santoro et al. (2017) | 81.07 | 98.82 | 89.95 |
| CNN+MLP Santoro et al. (2017) | 60.08 | 99.47 | 79.78 |
| Dense-Base | 46.93 | 57.71 | 52.32 |
| Dense-Small | 47.28 | 57.68 | 52.49 |

learn spatial relations across different samples and time steps contributing to task performance. Dense networks generally do not perform well in the more complex relational reasoning tasks.

## 6 CONCLUSIONS

We proposed the Associative Transformer (AiT), an architecture inspired by Global Workspace Theory and associative memory. AiT leverages a diverse set of priors with the emerging specialization property to enable enhanced association among representations via the Hopfield network. The comprehensive experiments demonstrate AiT's efficacy compared to conventional models, including the coordination method. In the future, we aim to investigate multi-modal competition within the shared workspace, enabling tasks to benefit from the cross-modal learning of distinct perceptual inputs.

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

## A APPENDIX

### A.1 DATASETS

In this section, we describe the datasets used in this work. (1) CIFAR-10 Krizhevsky (2009) is an image collection of 10 objects, covering 50k training samples and 10k test samples, labeled as airplane, automobile, and so on. The size of images is $32 \times 32 \times 3$. (2) CIFAR-100 Krizhevsky & Hinton (2009) contains 100 object classes with 500 training images and 100 testing images per class. For both the CIFAR-10 and CIFAR-100 datasets, we performed random cropping with size $32 \times 32 \times 3$ and a padding size of 4. (3) Triangle dataset Goyal et al. (2022b) includes 50k training images and 10k test images with size $64 \times 64$, each of which contains 3 randomly placed clusters of points. The task is to predict whether the three clusters form an equilateral triangle or not. (4) Oxford-IIIT Pet dataset Parkhi et al. (2012) comprises 37 categories featuring diverse breeds of cats and dogs, with 200 images allocated for each class. We utilized random resized cropping with size $256 \times 256 \times 3$ and resized all images to size $224 \times 224 \times 3$. Additionally, we applied random horizontal flip and normalization to the CIFAR-10, CIFAR-100, and Oxford-IIIT Pet datasets. (5) Sort-of-CLEVR dataset Santoro et al. (2017) is a simplified version of the CLEVR dataset Johnson et al. (2017). It includes 10k images with size $75 \times 75 \times 3$ and 20 different questions (10 relational and 10 non-relational questions) for each image. In each image, objects with randomly chosen shapes (square or circle) and randomly chosen colors (red, green, blue, orange, gray, yellow) are placed (Figure 6).

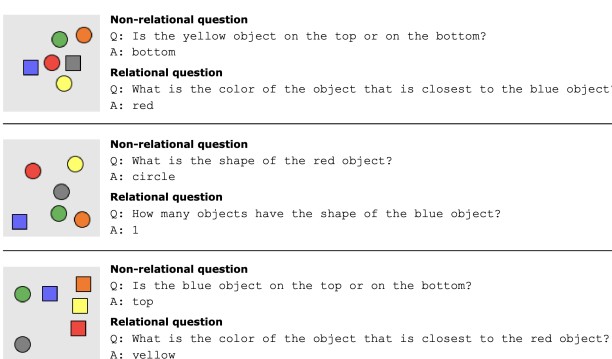

Figure 6: Examples from the Sort-of-CLEVR dataset Santoro et al. (2017).

### A.2 COMPARISON OF WORK RELATED TO GLOBAL WORKSPACE THEORY

We discuss and summarize the existing sparse attention methods in relation to the main properties of Global Workspace Theory (Table 5). First, we examine whether an architecture involves operations of information writing and reading through a shared workspace. Secondly, we assess whether the latent representations (priors) in workspace memory are subsequently processed by self-attention. Thirdly, we inspect whether the latent representations have a lower rank compared to the input representations. Fourthly, we analyze whether information retrieval from the workspace is driven by a bottom-up or a top-down signal. Lastly, we investigate whether the model incorporates a bottleneck with a limited capacity to regulate the information flow passing through the workspace.

Table 5: Comparison of attention architectures based on properties of the Global Workspace Theory

| Method | Operations | | Self-Attention | Low-Rank Memory | Top-Down/Bottom-Up | Bottleneck |
|---|---|---|---|---|---|---|
| | Writing | Reading | | | | |
| Vision Transformer Dosovitskiy et al. (2021) | - | - | - | - | BU | × |
| BlockBERT Qiu et al. (2020) | - | - | - | - | BU | ✓ |
| BRIMsMittal et al. (2020) | × | × | × | × | TD | ✓ |
| Modern Hopfield Ramsauer et al. (2021) | × | ✓ | × | ✓ | BU | × |
| Perceiver Jaegle et al. (2021) | ✓ | × | ✓ | ✓ | BU | × |
| Coordination Goyal et al. (2022b) | ✓ | ✓ | × | × | BU | ✓ |
| Perceiver IO Jaegle et al. (2022) | ✓ | ✓ | ✓ | ✓ | TD | × |
| Set Transformer Lee et al. (2019) | ✓ | ✓ | × | × | BU | × |
| Luna Ma et al. (2021) | ✓ | ✓ | × | × | BU | × |
| GMAT Gupta & Berant (2020) | ✓ | ✓ | ✓ | × | BU | ✓ |
| Associative Transformer (Ours) | ✓ | ✓ | × | ✓ | BU | ✓ |

## A.3 Experimental Settings and Hyperparameters

Table 6 presents the hyperparameters used for the different tasks in this study. Unless otherwise noted, we employed the author-recommended settings and hyperparameters for the re-implementation of baseline models. The small variants of ViT and AiT have 2 attention layers, and the base variants of them have 12 attention layers instead. We used the same dimension of the hidden layer and the MLP layer for ViT and AiT. By default, we employed 8 attention heads and 32 memory slots for the bottleneck attention. To obtain the bottleneck size, we considered two main factors of the batch size and the patch size. For the CIFAR and Pet datasets, we used a bottleneck size of 512, which selected from a pool of 32.8k/25.1k patches. For the Triangle dataset, we used a bottleneck size of 64 from a pool of 2.0k patches. For the relational reasoning tasks, we used a bottleneck size of 256, which selected from a pool of 14.4k patches. Based on the bottleneck size and the patch pool size, we used 128 memory slots for the Pet dataset and 32 memory slots for the other datasets. Moreover, we trained the models on the Pet dataset for 300 epochs and on the other datasets for 100 epochs. In relational reasoning tasks, we trained all models for 100 epochs with a batch size of 64.

Table 6: Hyperparameters

| Parameter | Value |
|---|---|
| **Common parameters** | |
| Optimizer | AdamW |
| Weight decay | 0.01 |
| Learning rate | $1 \times 10^{-4}$ |
| Number of self-attention heads | 12 |
| Number of attention layers | 2 (Small)/ 12 (Base) |
| Size of hidden layer | 768 |
| Size of MLP | 3072 |
| Size of memory slot | 32 |
| Number of bottleneck attention heads | 8 |
| Beta | 1.0 |
| Epochs | 100 (300 for Oxford Pet) |
| **CIFAR** | |
| Patch size | 4 |
| Batch size | 512 |
| Number of memory slots | 32 |
| Bottleneck size | 512 |
| **Triangle** | |
| Patch size | 32 |
| Batch size | 512 |
| Number of memory slots | 32 |
| Bottleneck size | 64 |
| **Oxford Pet** | |
| Patch size | 16 |
| Batch size | 128 |
| Number of memory slots | 128 |
| Bottleneck size | 512 |
| **Relational reasoning** | |
| Patch size | 5 |
| Batch size | 64 |
| Number of memory slots | 32 |
| Bottleneck size | 256 |

## A.4 Analysis of Operating Modes of Attention Heads

To understand the operating modes of attention heads in a vision Transformer (ViT) model, we measured the interaction sparsity of the pair-wise self-attention. The goal is to investigate whether sparse attention is required to learn meaningful representations during the model training. If the ViT model inherently learns such sparse interactions among patches, we can induce an inductive bias to foster the sparse selection of patches through a communication bottleneck. We trained a ViT-Base variant model for 100 epochs from scratch for the CIFAR-10 task. Then, for each attention head, we obtained a violin plot to represent the distribution of attention sparsity for different patches. The attention sparsity for a specific patch's interactions with other patches is computed as follows

$$\arg\min_s \sum_{j=1}^{s} A^{i,j} \geq 0.9, \tag{9}$$

where $A^{i,j}$ is the attention score allocated to the $j$th patch by the $i$th patch. The attention sparsity score is measured by the minimal number of required patches whose attention scores add up to 0.90. For instance, there are 65 patches for the CIFAR-10 task with patch size 4, thus there are 65 interactions for each patch to all the patches including itself. Then, an attention head has a higher sparsity if the median of the required patches $s$ that satisfies Equation 9 across different patches is smaller. Moreover, for each head, we showed the distribution of the attention sparsity scores for different patches in the violin plots (Figure 7), where the scores from different input samples were averaged. The number in the center of each panel gives the median $\bar{s}$ of the distribution. The heads in each layer are sorted according to $\bar{s}$. Note that training the model for a longer duration can result in even better convergence and higher attention sparsity. We also refer to a concurrent investigation on the attention sparsity of the Bidirectional Encoder Representations from Transformers (BERT) model for natural language processing (NLP) tasks Ramsauer et al. (2021). Our findings about the various operating modes of attention heads in ViT are in line with the findings in NLP tasks.

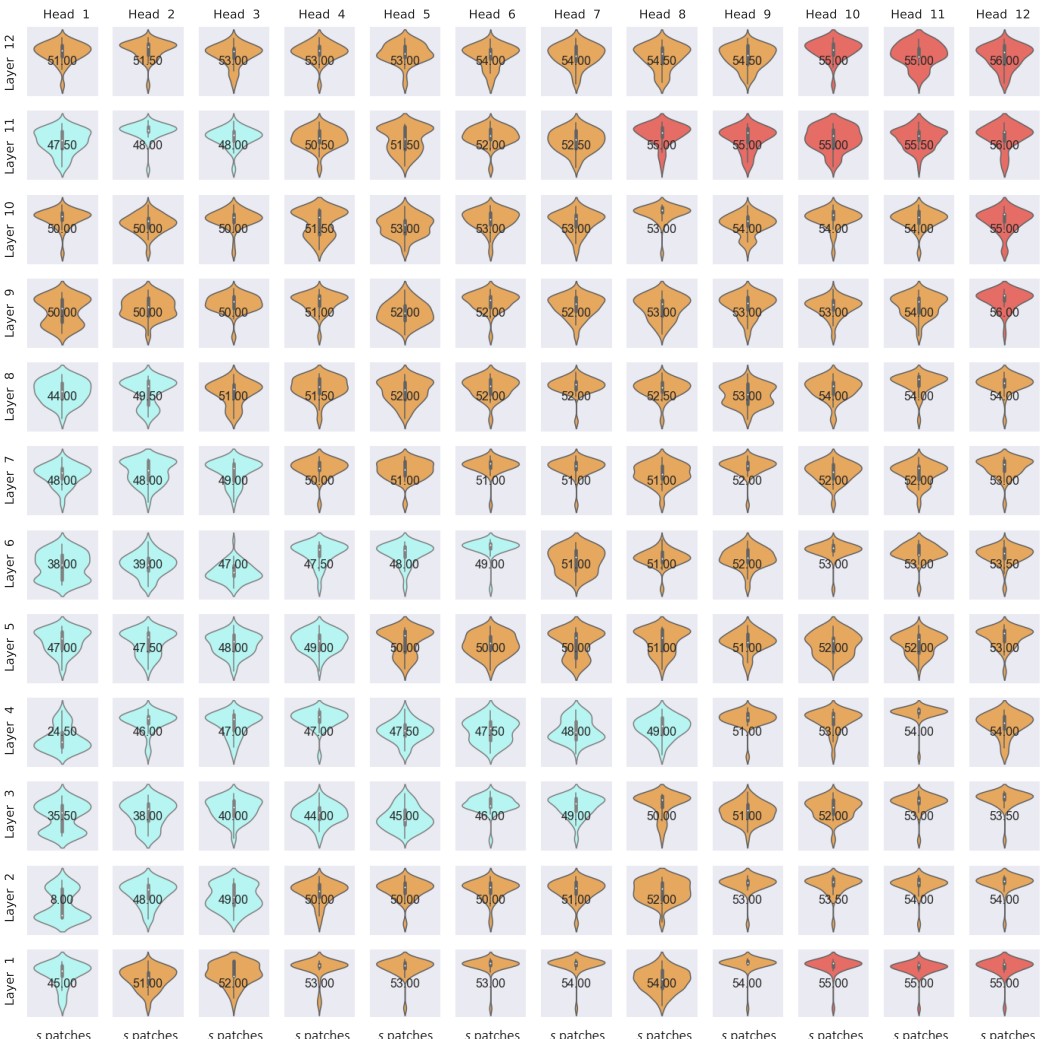

Figure 7: Analysis of operating modes of attention heads in the ViT-Base model. We recognize three different groups of attention heads based on their sparsity scores. Group (I) in light blue: High sparsity heads abundant in the middle layers 3-6. The vast majority of these heads only used 50% or fewer interactions. Group (II) in orange: Middle sparsity heads predominant in layers 2 and 7-10. Less than 80% of the interactions were activated. Group (III) in red: Low sparsity heads observed in high layers 11-12 and the first layer, where the most patches were attended to. The global workspace layer will provide the inductive bias to attend to the essential patches more effectively.

A.5 DISCUSSION ON THE TEST TIME SCHEME

We demonstrate the schemes for Associative Transformer during test time and the 'without memory forward' ablation in Figure 8. Moreover, we evaluated the model performance using the two different methods for both image classification and relational reasoning tasks (Table 7 and Table 8). The results show that AiT performs as well as or better than the 'without memory forward' ablation.

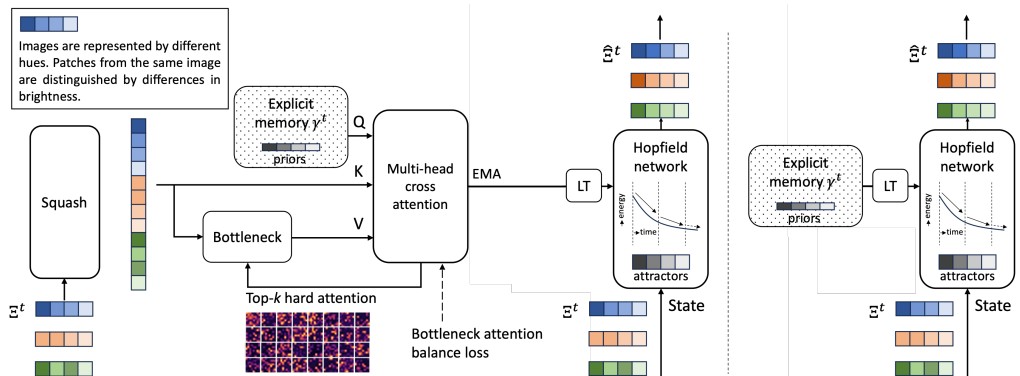

(a) Frozen memory during test time (AiT).

(b) Ablation without memory forward during test time.

Figure 8: The scheme of the Associative Transformer (AiT) during test time. (1) In AiT, although the explicit memory is frozen (depicted by filled dots), the memory forward process is enabled. However, the computed output of the multi-head cross attention will not be used to update the memory, which is different from training time. (2) In the 'without memory forward' ablation, the entire memory forward process is disabled, leveraging only Hopfield networks for retrieval.

Table 7: Image Classification Tasks

| Methods | CIFAR10 | CIFAR100 | Triangle | Pet |
|---|---|---|---|---|
| AiT-Medium | 84.59 | 60.58 | 99.57 | 30.05 |
| AiT-Medium (without memory forward) | 84.50 | 60.56 | 99.57 | 28.68 |

Table 8: Relational Reasoning Task

| Methods | Relational |
|---|---|
| AiT-Small | 76.82 |
| AiT-Small (without memory forward) | 75.17 |

A.6 EFFICACY OF THE BOTTLENECK ATTENTION BALANCE LOSS

The Bottleneck Attention Balance Loss facilitates the learning of priors that can attend to diverse sets of patches. We demonstrate the efficacy by visualizing the bottleneck attention scores computed using the learned priors (Figure 9) and the corresponding selected patches (Figure 10). We used as a metric for sparsity the ratio of distinct patches in all the selected patches by the bottleneck attention. With the progress of training, we can obtain a more diverse selection of patches.

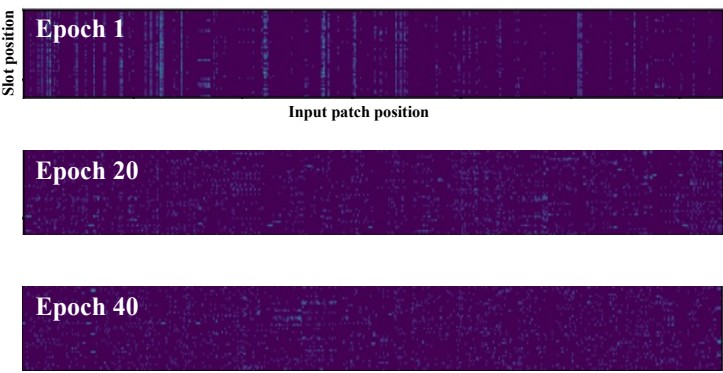

Figure 9: Bottleneck attention balance loss facilitates the selection of diverse patches from different input positions.

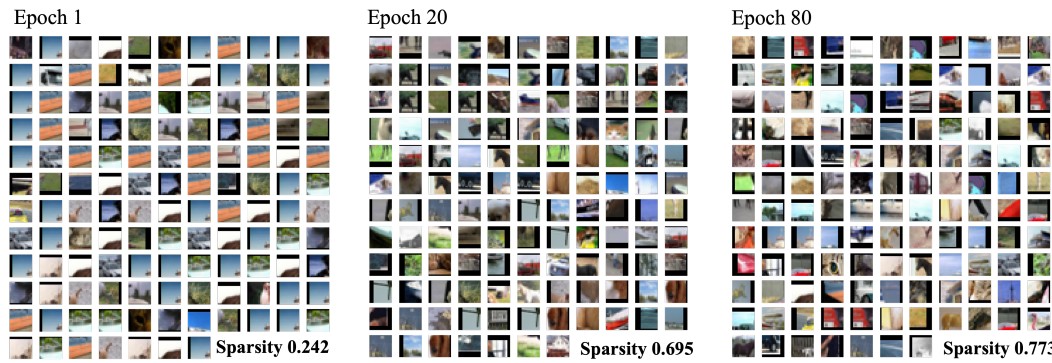

Figure 10: Examples of the selected patches by the bottleneck attention in CIFAR-10. As training progresses, we can obtain a more diverse selection of patches.

## A.7 HOPFIELD NETWORKS ENERGY

In traditional Hopfield networks, it is possible to store $N$ samples and retrieve them with partially observed or noisy patterns by updating model weights. During retrieval, these partially observed or noisy patterns converge to one of these attractors, minimizing the Hopfield energy. Unlike traditional Hopfield attractors that incorporate the implicit memory within its model parameters, AiT decouples the memory from the Hopfield network by introducing the learnable explicit memory. This memory serves the functions of both priors in the bottleneck attention and attractors in Hopfield networks. Consequently, the Hopfield network does not need to store different inner states every batch time, instead, we can reuse the learned memory bank from the bottleneck attention to update and maintain a set of attractors with the trainable linear transformation. The proposed architecture is an attractor network in the sense that, in every batch, a pattern converges to one of these attractors derived from the priors stored in the explicit memory bank.

Moreover, the information retrieval is based on a continuous Hopfield network, where an input state converges to a fixed attractor point within the associative memory of the Hopfield network. Usually, any input state that enters an attractor's basin of attraction will converge to that attractor. The convergence results in a decreased state energy with respect to the stored attractors in the memory. All patches reach their minimum at the same time and the global energy in Equation 7 is guaranteed to decrease. To quantitatively measure the amount of energy reduction during the information retrieval process in the Hopfield network, we computed an input state's energy before and after it was reconstructed. A successful retrieval results in substantial reduction in the state energy (Figure 11).

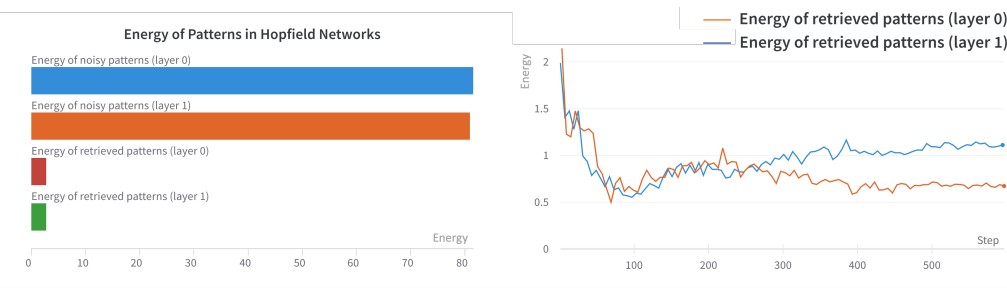

(a) Global energy over the implicit iteration time step

(b) Global energy over the explicit batch time steps

Figure 11: Energy of patch representations in the CIFAR-10 task for AiT-Small. The Hopfield network operates by iteratively decreasing the energy of an input state with respect to the attractors stored in its memory. This reduction in energy enables the retrieval of a representation that closely aligns with learned attractors, effectively leveraging knowledge within the associative memory. (a) The global energy is guaranteed to decrease over the implicit iteration time step for every retrieval. (b) The global energy over the explicit batch time steps generally decreases, especially during the early stages of training, since the computed energy is the relative energy to the memory bank that is updated every batch time step.

