# OpenReview forum: "Associative Transformer is a Sparse Representation Learner"
_ICLR.cc/2024/Conference — Submitted to ICLR 2024_

### Official Review · Reviewer_7hau · 2023-11-01

**Soundness:** 2 fair
**Presentation:** 2 fair
**Contribution:** 2 fair
**Rating:** 6
**Confidence:** 3

**Summary:**

The authors propose the Associative Transformer (AiT) — a special Transformer block that supplements the normal (Attention, MLP) Transformer Blocks that augments the Transformer with associative memory. This special layer allows tokens from different samples in a batch to interact with each other via a modified attention operation. The model improves the baseline set by the original ViT at the cost of a few extra parameters.

**Strengths:**

## Novel and reasonably well-defended
- (+) The ablation study shows that we can expect a consistent ~2% performance gain on image classification tasks by including the GWT sub-layer into a Transformer Block.
- (+) The computational complexity of using more tokens in the attention is mitigated by an intelligent use of "hard attention" to select only the most meaningful tokens in the squashed input.
- (+) To my knowledge, this supplemental memory sub-block is novel.

**Weaknesses:**

## Paper lacks clarity in some areas

1. (- -) Allowing samples across batches to compete via the bottleneck attention is inelegant -- inherently, this competition means that the ViT's prediction on an input is dependent on what samples are in the same batch. What is the performance if we use batch size = 1 during evaluation? What about different batch sizes? Unless I am misunderstanding something, it would be necessary to report standard deviation across different evaluation batch sizes when reporting metrics for this kind of network.
2. (-) I do not find the attention maps in Fig 2 meaningful. Which layer's memory bank was used to display this attention map? How is the attention heatmap smooth if it operates on patches (not pixels)? These questions are not answered in the paper.
3. (-) I suspect there are many more FLOPS needed for this network if the Hopfield Network component of the GWT sub-block employs recurrence. If true, this should be reported and quantified as a limitation of the model.
4. (-) Fig. 1 is very unclear to me. See questions.

**Questions:**

I have a few confusions the authors could help clarify, many related to the display and caption of Fig. 1.

1. Caption of Fig 1 says: "...squashed into vectors $\mathbb{R}^{(B \times N) \times E}$". Is this not a matrix, where you combine the batch and token dimension? Perhaps this means a collection of vectors, but then the figure itself is confusing because we see only one vector passed into the bottleneck attention.
2. "MASK" is not defined in Fig 1
3. In Fig 1, there are two blocks of "Explicit Memory" -- do these represent the same matrix?
4. In Fig 1, why are there three inputs to the Hopfield Network `(Explicit Memory, LT(explicit memory), state)`? Where is the update equation for the Hopfield Network?

---

> ### Author Response · Authors · 2023-11-12
> **Author feedback to Reviewer 7hau [1/2]**
>
> We are sincerely grateful for the precious review time and valuable comments.
>
> **Q1: Allowing samples across batches to compete via the bottleneck attention is inelegant.**
>
> ViT's prediction for an input is not solely dependent on the samples within the same batch. To elaborate on this, the memory bank update utilizes Exponential Moving Average (EMA) with a smoothing factor $\alpha = 0.9$ (that determines the rate at which older observations decay) in Equation 4, accumulating information from both old $\gamma^{t}$ and new memories $\hat{\gamma}^{t}$.
>
> $\gamma^{t+1} = \alpha \cdot \gamma^{t} + (1 - \alpha) \cdot \hat{\gamma}^{t}$.
>
> This memory update strategy with a high smoothing factor $\alpha = 0.9$ based on EMA ensures the stable performance of the proposed architecture with varying batch sizes. The learned explicit memory encompass not only information from the current batch but also from all batches observed thus far. Therefore, given that the overall training data distribution remains constant, using different batch sizes is not expected to significantly impact model behavior.
>
> The only factor that might affect the model's behavior concerning batch size is that, intuitively, a larger batch size may introduce stronger competition among patches with a constant bottleneck size. In our experiments, we employed grid search to tune the patch size, batch size, and bottleneck size. In P13 Appendix A.3 "Experimental Settings and Hyperparameters", we provided detailed settings.
>
> We aimed to maintain a ratio between the bottleneck size and the total patch numbers, as shown to be effective in previous studies on the Coordination method. For the CIFAR and Pet datasets, we used a bottleneck size of 512, selected from a pool of 32.8k/25.1k patches, respectively. For the Triangle dataset, we used a bottleneck size of 64 from a pool of 2.0k patches. For the relational reasoning tasks, we used a bottleneck size of 256, selected from a pool of 14.4k patches. The difference with the Coordination method is that we introduced the competition among different samples. This allows a sample's classification task, for instance, to benefit from other patches within the batch input.
>
> When using a batch size of 1 during evaluation, the proposed bottleneck attention with the squash layer degrades to the traditional sample-wise bottleneck in the Coordination method. We could adjust the bottleneck size for evaluation to a smaller value, for example 1, when the task is CIFAR10 with a patch size of 4. Alternatively, using a fixed memory learned during training for evaluation is possible. The memory remains static during evaluation, and the batch is directly sent to the Hopfield network for reconstruction based on the fixed memory.
>
> **Q2:  I do not find the attention maps in Fig 2 meaningful. Which layer's memory bank was used to display this attention map? How is the attention heatmap smooth if it operates on patches (not pixels)? These questions are not answered in the paper.**
>
> Thank you for pointing this out. The attention map pertains to the memory bank of the first layer, representing the attention scores across different **patch** locations (8 rows and 8 columns, totaling 64 locations, with one pixel for each). The attention maps are computed by averaging the attention scores across different samples, resulting in the smooth visualization shown in Figure 2.
>
> **Q3: I suspect there are many more FLOPS needed for this network if the Hopfield Network component of the GWT sub-block employs recurrence. If true, this should be reported and quantified as a limitation of the model.**
>
> Regarding the functioning of the Hopfield network, it does not employ recurrence during the end-to-end training and features a decoupled memory (the explicit memory bank) for efficient training.
>
> In detail, Equation 7 describes an iteration that could be applied multiple times, but typically, we use just a single step for forward and backward computation during end-to-end training. Here, $t$ represents the batch time step, and the iteration time step is implicit. The Hopfield network utilizes a single forward step to recall patterns in the memory bank. In every retrieval, the global energy is guaranteed to decrease over time step(s), as illustrated in Figure 10.
>
> Moreover, the Hopfield network does not need to compute and store inner states for every batch input, which could increase the FLOPS. Instead, we utilize the explicit memory bank (Figure 1.a) for the update of memory slots. Alternatively, the Hopfield network can also incorporate an implicit memory by introducing trainable parameters. In this case, the patterns are stored in the weights of these parameters requiring more FLOPS. Differing from the implicit memory, in the proposed architecture, we employ the same memory architecture for learning both the priors for the bottleneck attention and the attractors for the associative memory in the Hopfield network.

---

> ### Author Response · Authors · 2023-11-12
> **Author feedback to Reviewer 7hau [2/2]**
>
> **Regarding the Figure 1**
>
> **Q4.1: Caption of Fig 1 says: "...squashed into vectors $\mathbb{R}^{(B\times N)\times E}$". Is this not a matrix, where you combine the batch and token dimension? Perhaps this means a collection of vectors, but then the figure itself is confusing because we see only one vector passed into the bottleneck attention.**
>
> Yes, it is a collection of vectors. We revised the Figure 1 where images are represented by different colors. Patches from the same image are distinguished by a gradient with each square representing the vector $\mathbb{R}^{E}$ of a patch.
>
> **Q4.2: "MASK" is not defined in Fig 1**
>
> The "Mask" in Figure 1 was the top-$k$ hard attention. We revised the figure and replaced the Mask accordingly.
>
> **Q4.3: In Fig 1, there are two blocks of "Explicit Memory" -- do these represent the same matrix?**
>
> Yes, the two blocks of "Explicit Memory" represent the same matrix, but at different time steps. The first explicit memory in Figure 1 corresponds to the memory slots $\gamma^t$ at the time step $t$, while the second explicit memory represents the updated memory $\gamma^{t+1}$ based on the bottleneck attention. The illustration of the second explicit memory block is intended to demonstrate that the updated memory $\gamma^{t+1}$ is employed as the input to the Hopfield network.
>
> **Q4.4: In Fig 1, why are there three inputs to the Hopfield Network (Explicit Memory, LT(explicit memory), state)? Where is the update equation for the Hopfield Network?**
>
> Thank you for pointing this out! The inputs to the Hopfield network are twofold, instead of the three that were shown in Figure 1. We revised the figure 1 and the two inputs are the output of the linear transformation of the explicit memory $f_{\text{LT}}(\gamma^{t+1})$, and the current patch representations (the input state) $\Xi^t$.
>
> The update equation of the Hopfield network is based on the modern Hopfield proposed in [1]. The update rule for a patch representation $\xi^t \in \Xi^t$ in the Hopfield network is $\hat{\xi}^t=f_{\text{LT}}(\gamma^{t+1}) \text{softmax}(\beta (f_{\text{LT}}(\gamma^{t+1}))^T \xi^t)$, which can be derived from Equation 7. A skip connection is employed to obtain the final output $\Xi^{t+1}=\hat{\Xi}^t+\Xi^t$.
>
> [1] Hubert Ramsauer, Bernhard Schafl, Johannes Lehner, and et al. Hopfield networks is all you need. ICLR 2021.

---

> > ### Comment · Reviewer_7hau · 2023-11-22
> >
> > I thank the authors for their clarifications and update to the manuscript (in particular, the clarifications to Fig 1 are extremely helpful). However, after re-reading the revised manuscript I believe my original score of a 6 still reflects my current assessment of this work.

---

### Official Review · Reviewer_vNUE · 2023-11-04

**Soundness:** 3 good
**Presentation:** 3 good
**Contribution:** 2 fair
**Rating:** 3
**Confidence:** 3

**Summary:**

This work proposes a new neural network model called Associative Transformer (AiT) that uses a sparse attention mechanism to process input data based on biological principles. The model divides the input data into subsets and associates each subset with others to share information between them. The experimental results show that the AiT model outperforms traditional Transformer models and other methods in multiple vision tasks.

**Strengths:**

The model is novel inspired by human brain's associative memory.

**Weaknesses:**

1. The paper does not include any ablation studies.
2. The experimental evaluation of the model is limited to a few vision datasets.
3. The work does not provide the available code.

**Questions:**

1. How does the model's performance vary when different types of priors are used in the workspace memory?
2. How does the model's performance vary when different types of attention mechanisms are used?
3. What are the limitations of the proposed model?
4. The baseline seems not include SOTA models in vision tasks.

---

> ### Author Response · Authors · 2023-11-12
> **Author feedback to Reviewer vNUE [1/2]**
>
> We are sincerely grateful for the precious review time and valuable comments.
>
> **Q1: Ablation studies**
>
> We performed several ablation studies in P7 Section 5.3 Ablation study and P8 Section 5.4 Comparison with the coordination method. In particular, the Table 2 "Comparison based on an ablation study" and the Figure 4 "Model size vs. accuracy for configurations" were discussed in details.
>
> | Models               | CIFAR10 | CIFAR100 | Triangle | Average   |
> |----------------------|---------|----------|----------|-----------|
> | AiT                  | **83.34** | **56.30** | **99.47** | **79.70** |
> | Reset memory         | 81.94 | 55.96 | 99.46 | 79.12 |
> | W/O Hopfield         | 81.03   | 54.96    | 99.44    | 78.48     |
> | W/O memory (ViT)     | 79.53   | 53.19    | 99.47 | 77.40 |
> | Dense networks       | 77.78   | 53.14    | 99.46    | 76.79     |
> | W/O bottleneck       | 75.40   | 46.53    | 93.33    | 73.75     |
> | W/O SA               | 72.72   | 47.75    | 99.46    | 73.31     |
> | W/O FF               | 69.51   | 40.89    | 97.61    | 69.34     |
>
> **Q2: Datasets**
>
> Following the previous studies, we experimented on the datasets of the Triangle, CIFAR10, CIFAR100, Oxford-IIIT Pet, and Sort-of-CLEVR. The detailed explanation of these datasets was clarified in P12 Appendix A.1 Datasets.
>
> **Q3: Code**
>
> As discussed in P6 Section 5.1 Experiment Setup, the code for the implementation of the AiT architecture and the training on all the experimental datasets will be made publicly available.
>
> **Q4: How does the model's performance vary when different types of priors are used in the workspace memory?**
>
> Thank you for pointing this out! All priors are used during the training however with different expertise to process different patches. In Figure 2, we visualize the attention maps generated by each prior. These maps highlight the same specific locations across all input samples during the selection of relevant image patches for different priors.
>
> Additionally, in P15 Figure 9, the examples of the selected image patches by the prior-guided attention are visualized. As training progresses, a more diverse selection of patches can be obtained.
>
> **Q5: How does the model's performance vary when different types of attention mechanisms are used?**
>
> We provided a comprehensive evaluation of the proposed attention mechanism, both in comparison to existing models, and through internal performance analysis in the context of ablation studies. First, the comparison between the proposed global workspace layer (Figure 1.a) and conventional sparse attention-based workspace methods is shown in Table 1, including the Coordination, Perceiver, Set Transformer, and Luna methods. Moreover, we conducted an ablation study regarding the attention mechanism to understand how the model's performance varies when different types of attention mechanisms are used. In Table 2, under the row labeled "W/O Hopfield," we measured the model's performance when replacing the Hopfield network with another cross-attention mechanism. The added cross-attention has the same architecture as the multi-head attention in Figure 1.a, but without applying the bottleneck attention. Unlike the multi-head attention in the global workspace layer, the added cross-attention in the ablation study takes the current input state as the query, using the contents in the explicit memory as the keys and values. The output of the cross-attention is used as the output of this layer. The ablation study in Table 2 demonstrated the efficacy of the proposed method by investigating different attention mechanisms of the cascaded cross-attention and the self-attention components.
>
> **Q6: What are the limitations of the proposed model?**
>
> We discussed several limitations in Section 9. First, the intricacies of managing multiple modalities simultaneously pose a current challenge in the methodology, although AiT demonstrates superior performance in its capability to attend to and retrieve resembling patches from the shared workspace (please refer to Section 5.8 Relational Reasoning). We aim to investigate multi-modal competition within the shared workspace, enabling tasks to benefit from the cross-modal learning of distinct perceptual inputs. For example, when the input vectors are from both image patches and text tokens, the bottleneck attention could adaptively select vectors from different modalities and learn cross-modal priors that guide attention.
>
> Additionally, in Section 5.7, we investigated the effect of the inverse temperature, $\beta$, on information retrieval based on the Hopfield networks in Figure 5. The limitation arises in the need for tuning the $\beta$ value to reach optimal performance. We aim to study a mechanism to adjust the $\beta$ adaptively in the future, addressing this limitation and potentially further improving performance.

---

> ### Author Response · Authors · 2023-11-12
> **Author feedback to Reviewer vNUE [2/2]**
>
> **Q7: The baseline seems not include SOTA models in vision tasks.**
>
> A crucial aspect of this study is its exclusive comparison of models that exhibit most of the properties outlined in the Global Workspace Theory. In Appendix A.2 'Comparison of Work Related to Global Workspace Theory', Table 5 demonstrates the comparison of attention architectures based on the properties of the Global Workspace Theory and how existing studies are related to these properties. The Vision Transformer (the first row) serves as an example to illustrate its inapplicability to implement the shared workspace. Therefore, other related conventional Transformer models typically do not fall within the scope of sparse attention-based workspace approaches.
>
> The SOTA method in the context of sparse attention workspace-based approaches is the Coordination method by Goyal et al. (2022) [1], where a shared coordination space was proposed to correlate different information fragments. To investigate further, we conducted an additional ablation study in Section 5.4. The detailed comparison with the coordination method is presented in Figure 4, where we showed that integrating all three components (C+LM+HN+SA) resulted in a competitive accuracy of 71.49\% with a compact model size of only 1.0M. These results highlight that the proposed AiT is an efficient sparse representation learner, outperforming the existing SOTA method of Coordination. Additional models compared in this study include the Set Transformer, Luna, Perceiver, and BRIMs, all of which exhibit most of the properties outlined in the Global Workspace Theory. We are pleased to provide clarification if any points may be unclear or if additional results are necessary for understanding the AiT architecture and its contributions to the existing literature.
>
> [1] Anirudh Goyal, Aniket Rajiv Didolkar, Alex Lamb, and et al. Coordination among neural modules through a shared global workspace. ICLR 2022.

---

### Official Review · Reviewer_H5y7 · 2023-11-09

**Soundness:** 3 good
**Presentation:** 3 good
**Contribution:** 2 fair
**Rating:** 3
**Confidence:** 3

**Summary:**

The authors propose the Associative Transformer (AiT), inspired by global workspace theory and (Modern) Hopfield networks.  They introduce a new module called a global workspace layer (GWL) that can be stacked after a standard Vision Transformer (ViT) module and demonstrate that their module can lead to modest improvements in image classification tasks compared to ViT and greater gains compared to other models, some of which however do better than AiT on relational reasoning tasks. They perform several ablations that help delineate which aspects of their model contribute the most to its success, and include one experiment that suggests their module can produce an enhancement of performance relative to a ViT without their added module.

**Strengths:**

The proposed GWL introduces several rich properties into the computation of visual representations, and I found myself feeling that it was useful to explore the properties of these innovations.

The ablation experiments helped clarify the roles of different features of the GWL, and the experiment comparing AiT to more vanilla ViT modules suggested a possible advantage of the model that would be worth more fully understanding in future work.  The comparisons with the coordination method also introduced some potentially interesting advantages of the method on the CIFAR 10 dataset.

**Weaknesses:**

As a general reaction, this seems like exploratory work, and it's possible that the next iteration will be a big improvement.  I hope the following comments are food for thought in case others agree with my assessment that this iteration is not above threshold for acceptance.

For me the most important limitation of the work is the overall complexity of the GWL together with the relatively small advantage it offered compared to the standard ViT and even to compared to simple Feed-forward neural networks (called Dense in the ablation study).  Without a bigger advantage over these systems, and with the shortcomings of AiT on relational reasoning, it is hard to see clear evidence of an advantage for AiT over existing methods, especially given its complexity.

Although an experiment with the Pet dataset did suggest that with larger data sets the advantage of AiT could be larger, performance of both models was very poor, and if I understand correctly the AiT had far more parameters (there was no compensation for the added parameters in the GWL's added to the model).  The only place where I saw a consideration of parameters was in the exploration of integrating features of the GWL into variants of the coordination model.

In general, it seems to be a weakness that the GWL is an add, not a replacement, for the ViT module.

I also had difficulty understanding several aspects of the model, which I ask about under questions below.

**Questions:**

Responses to these questions or strong rebuttals to the weaknesses I've raised could potentially increase my rating.  In any case I'd be interested in seeing responses to make sure I've understood.

I was uncertain about the functioning of the Hopfield net.  Do these equations describe an iteration that is applied some number of times?  If I'm correct, the t is not the iteration time step -- it is the batch time step -- but perhaps an iteration time step is implicit. In that case, the expression for ˆξt in Eq (8) right, could be understood as the minimum of ξt over these implicit time steps.  Is all this correct?  If not, please explain.  If so, why not just use the final time value at the end of the settling process?  Are these values reaching their minimum at different times for different patches?  Is the global energy Ξt guaranteed to decrease over time steps?  If not, in what sense is this really an attractor network?

In introducing the GWL, the text states: "The global workspace layer can be seen as an add-on component on the monolithic Vision Transformer, where the feed-forward layers process patches before they enter the workspace, facilitating abstract relation learning, and the self-attention learns the contextual relations for a specific sample. The global workspace layer learns spatial relations across various samples and time steps."  This text made me think that the initial hope was that the model would actually improve relational reasoning.  It did not, but this was attributed to 'the difficulty in accurately reconstructing question representations from the memory'.  This is testable by using the AiT and comparison models to derive and image representation for use as a component of a combined model that gets its embedding of the question from a separate language-processing module.  If you have results from such an experiment, it would be interesting to hear about them.

In the ablation studies, I was uncertain how to interpret the W/0 Hopfield ablation.  The text says "W/O Hopfield evaluates the model performance when we replace the Hopfield network with another cross-attention mechanism."  I need a much more explicit understanding of this.  What exactly is being replaced with what.  Also, please clarify the meaning of 'W/O SA examines performance when the self-attention
component is excluded'.  Is the self-attention component the multi-head attention block in fig 1b?  This led to major decrements, but there was no mention of this fact or its meaning for our understanding of the proposed architecture.

Can you comment on my sense that it is a weakness of the paper that the GWL is an add, rather than a replacement, to the ViT module?  Would one way to balance parameters be to reduce the total stack depth of the ViT when adding GWLs, as these are really added layers?  Have you tried that?

Demonstrations of bigger advantages that might have been obtained since submission could potentially cause me to rate this paper higher.

Please address the issue of the lack of control for number of parameters.

---

> ### Author Response · Authors · 2023-11-16
> **Author feedback to Reviewer H5y7 [1/3]**
>
> We are sincerely grateful for the precious review time and valuable comments.
>
> **Q1: For me the most important limitation of the work is the overall complexity of the GWL together with the relatively small advantage it offered compared to the standard ViT and even to compared to simple Feed-forward neural networks (called Dense in the ablation study). Without a bigger advantage over these systems, and with the shortcomings of AiT on relational reasoning, it is hard to see clear evidence of an advantage for AiT over existing methods, especially given its complexity.**
>
> Thank you very much for pointing this out. We have added further explanation with additional experiments to demonstrate the benefits of the GWL:
>
> 1. We introduced a new variant of the AiT model with a size of 45.9M and 6 attention layers. This model outperforms the larger ViT-Base model, which has a size of 85.7M and 12 attention layers.
>
> 2. We improved the performance on both the Pet dataset and the Relational Reasoning dataset for the proposed AiT, as shown in Figure 3 and Table 4. The results demonstrate a substantial improvement in the performance of AiT.
>
> 3. We conducted experiments demonstrating that dense networks cannot achieve good performance in more complex relational reasoning tasks, despite their relatively high accuracy in ablation studies (Table 4).
>
> We provide below a detailed explanation of these points by addressing your following questions.
>
> **Q2: I was uncertain about the functioning of the Hopfield net. Do these equations describe an iteration that is applied some number of times? If I'm correct, the $t$ is not the iteration time step -- it is the batch time step -- but perhaps an iteration time step is implicit. In that case, the expression for $\hat{\xi}^t$ in Eq (8) right, could be understood as the minimum of $\xi^t$ over these implicit time steps. Is all this correct? If not, please explain. If so, why not just use the final time value at the end of the settling process? Are these values reaching their minimum at different times for different patches? Is the global energy $\Xi^t$ guaranteed to decrease over time steps? If not, in what sense is this really an attractor network?**
>
> Regarding the functioning of the Hopfield network, your understanding is correct. The equations describe an iteration in Hopfield network that can be applied multiple times. Usually, we apply only a single interaction for the efficiency of end-to-end training, as recurrence can disrupt parallel processing. Therefore, in Equation 8, $t$ represents the batch time step, and the iteration time step is implicit. A single forward step is used to reconstruct patterns based on priors in the memory bank, obtaining the final time value at the end of the settling process.
>
> Moreover, in the modern Hopfield network, all patches reach their minimum at the same time. The global energy is guaranteed to decrease over the implicit iteration time step for every retrieval, as depicted in Figure 10.a. The global energy over the explicit batch time steps generally decreases, especially during the early stages of training, since the computed energy is the relative energy to the memory bank that is updated every batch time step. An example of the energy trend over the batch time steps is shown in Figure 10.b.
>
> To be more concrete, we explain why using an explicit memory bank and its relation to attractors. In traditional Hopfield networks, we store $N$ samples and retrieve them with noisy patterns by updating model weights. During retrieval, these noisy patterns converge to one of these attractors, minimizing the Hopfield energy. Unlike traditional Hopfield attractors that incorporate the implicit memory within its model parameters, the proposed AiT decouples the memory from the Hopfield network by inducing the learnable explicit memory bank. Consequently, the Hopfield network does not need to update inner states every batch time, instead, we can reuse the learned memory bank from the bottleneck attention mechanism to maintain a set of attractors. It is an attractor network in the sense that, in every batch, a pattern converges to one of these attractors that are derived from the priors in the explicit memory. This memory bank serves the functions of both priors in the bottleneck attention and attractors in the Hopfield network.

---

> ### Author Response · Authors · 2023-11-16
> **Author feedback to Reviewer H5y7 [2/3]**
>
> **Q3: In introducing the GWL, the text states: "The global workspace layer can be seen as an add-on component on the monolithic Vision Transformer, where the feed-forward layers process patches before they enter the workspace, facilitating abstract relation learning, and the self-attention learns the contextual relations for a specific sample. The global workspace layer learns spatial relations across various samples and time steps." This text made me think that the initial hope was that the model would actually improve relational reasoning. It did not, but this was attributed to 'the difficulty in accurately reconstructing question representations from the memory'. This is testable by using AiT and comparison models to derive and image representation for use as a component of a combined model that gets its embedding of the question from a separate language-processing module. If you have results from such an experiment, it would be interesting to hear about them.**
>
> Thank you very much for your suggestion. We have reimplemented the experiments on relational reasoning tasks, leveraging a text embedding network that comprises a learnable linear projection layer and layer normalization before and after the linear projection. Results across three different seeds demonstrate a substantial improvement in AiT's performance when addressing relational reasoning tasks compared to the coordination method. Additionally, AiT achieved a competitive performance of 88.34\% with other non-Transformer-based models, including the best-performing CNN+RN method with a performance of 89.95\%. The non-Transformer based models incorporate inductive biases into their architectures, which often results in superior performance compared to the Transformer based methods that lack a built-in inductive bias. This suggests that the global workspace layer can learn spatial relations across different samples and time steps, contributing to the enhanced performance.
>
> ### **Relational reasoning tasks**
> | Methods               | Relational | Non-relational | Average |
> |-----------------------|------------|----------------|---------|
> | **Transformer based models** |            |                |         |
> | AiT-Small             | **76.82**  | **99.85**      | **88.34**|
> | Coordination  | 73.43 | 96.31          | 84.87   |
> | Set Transformer | 47.63 | 57.65          | 52.64   |
> | **Non-Transformer based models** |        |                |         |
> | CNN+RN | 81.07      | 98.82          | 89.95   |
> | CNN+MLP   | 60.08      | 99.47          | 79.78   |
> | Dense-Base            | 46.93      | 57.71          | 52.32   |
> | Dense-Small           | 47.28      | 57.68          | 52.49   |
>
> In comparison, despite the relatively high accuracy of dense networks in the ablation task in Table 2, their performance in relational reasoning tasks is considerably inferior to AiT. Notably, we investigated the Dense-Small network with 11.1M parameters and the Dense-Base network with 62.7M parameters. The Dense-Small achieved 52.49\%, and the Dense-Base achieved 52.32\% in relational reasoning tasks. In contrast, the AiT-Small with 15.8M parameters achieved 88.34\% in relational reasoning tasks. We have included a discussion related to the dense network's performance in Section 5.8 Relational Reasoning.
>
> **Q4: In the ablation studies, I was uncertain how to interpret the W/O Hopfield ablation. The text says "W/O Hopfield evaluates the model performance when we replace the Hopfield network with another cross-attention mechanism." I need a much more explicit understanding of this. What exactly is being replaced with what. Also, please clarify the meaning of 'W/O SA examines performance when the self-attention component is excluded'. Is the self-attention component the multi-head attention block in fig 1b? This led to major decrements, but there was no mention of this fact or its meaning for our understanding of the proposed architecture.**
>
> Thank you for pointing this out. In the "W/O Hopfield" ablation, the Hopfield network is substituted with another multi-head attention (MHA) that shares the same architecture as the self attention in Figure 1.b. The rationale is grounded in the prior studies of Set Transformer and Perceiver models that relied on two MHA components cascaded in depth. For a fair comparison, instead of simply removing the Hopfield network, we replaced it with the MHA in the ablation study. The added MHA takes the input state $\Xi^t$ as the query and the upscaled priors in the explicit memory $f_{\text{LT}}(\gamma^{t+1})$ as the key and value, i.e., $\hat{\Xi}^{t} = \text{MHA}(\Xi^t,f_{\text{LT}}(\gamma^{t+1}))$.
>
> In the "W/O SA" ablation, the term "SA" refers to the multi-head self attention block in Figure 1.b. We have explicitly clarified this point in Section 5.3 Ablation Study. Additionally, we have revised Figure 1 to distinctly differentiate between the multi-head cross attention in Figure 1.a and the multi-head self attention in Figure 1.b.

---

> ### Author Response · Authors · 2023-11-16
> **Author feedback to Reviewer H5y7 [3/3]**
>
> **Q5: Can you comment on my sense that it is a weakness of the paper that the GWL is an add, rather than a replacement, to the ViT module? Would one way to balance parameters be to reduce the total stack depth of the ViT when adding GWLs, as these are really added layers? Have you tried that?**
>
> Thank you very much for your suggestion. We have introduced a new AiT model, "AiT-Medium", featuring 6 layers with 45.9M parameters, a shallower and smaller network compared to the ViT-Base (12 layers with 85.7M parameters). We added experiments with AiT-Medium on the CIFAR10, CIFAR100, Triangle, and Pet datasets. The results in Table 1 demonstrate that the AiT-Medium exhibited an average performance of 81.58\% for the CIFAR and Triangle datasets, surpassing the ViT-Base model's performance of 80.46\% while requiring significantly fewer parameters. Furthermore, in the Pet dataset, we conducted a comparison among AiT-Small, AiT-Medium, ViT-Small, and ViT-Base. Figure 3 illustrates that AiT-Medium also outperforms ViT-Base in the Pet dataset. AiT converges at a later stage of model training, due to its smaller size with fewer layers to compute per batch compared to ViT-Base.
>
> ### **Performance comparison in image classification tasks**
> | Methods             | CIFAR10 | CIFAR100 | Triangle | Average | Model Size |
> |---------------------|---------|----------|----------|---------|------------|
> | AiT-Base            | **85.44** | **60.78** | 99.59 | **81.94** | 91.0       |
> | **AiT-Medium**          | 84.59   | 60.58    | 99.57   | 81.58   | 45.9       |
> | AiT-Small           | 83.34   | 56.30    | 99.47   | 79.70   | 15.8       |
> | | | | | | |
> | **ViT-Base**  | 83.82   | 57.92    | **99.63** | 80.46 | 85.7       |
> | ViT-Small           | 79.53   | 53.19    | 99.47   | 77.40   | 14.9       |
>
> Therefore, we believe that the GWL add-on provides substantial advantages, as evidenced by the improved performance resulting from the reduced total stack depth of the model. The additive nature of GWL also facilitates efficient implementation across various applications. In our experiments, we thoroughly investigated its benefits in enhancing conventional models, including both coordination methods and Vision Transformers, demonstrating its applicability and potential for a broader range of use cases.

---

> ### Comment · Reviewer_H5y7 · 2023-11-21
> **Improved paper, further suggestion for final improvement for clarity**
>
> I think the authors have done a lot to address my questions and comments. The additional experiments related to relational reasoning and the introduction of the AiT-Medium model add useful comparisons supporting the value of the overall proposal.  I feel that I understand the paper better now, and I think it would be useful for others to hear about this work, so I encourage acceptance, and am moving up to marginal accept.
>
> As I read the paper again, I feel that I am beginning to understand it but that it is still not easy to understand.  I think a big part of this is that it is difficult for the reader to track what is happening at the batch level and what is happening as each item is being processed.  As I understand it now, everything in the flgure up to Explicit memory \gamma^{t+1} is taking a batch and returning a small set of updated priors.  These priors are then used to pull the representation of each image in the direction of the attractors represented by the priors.  Crucially, If I understand correctly, all the attractors act on each image to adjust it in ways that enhance performance on the task.  I think figure 1 could be drawn in a way that makes this clearer, perhaps with Ξt at the left being processed through an upper track, with the Hopfield network receiving the lower track and emitting ˆΞt sending this forward. The caption should emphasize that the same small set of 32 upscaled attractors are used to adapt each image.
>
> Also, I wondered if, at test time, it would make sense to freeze the explicit memory so that it is not being adjusted at all within a batch.  Maybe that's what is already happening?  In any case, this would dissociate the idea that these are priors with the idea that somehow they are being induced from the batch, and would make a useful ablation.  I'd apprecate a comment on this before I finalize my score.

---

> > ### Author Response · Authors · 2023-11-22
> > **Author feedback to Reviewer H5y7**
> >
> > We are sincerely grateful for the valuable comments and positive feedback.
> >
> > **Q1: As I read the paper again, I feel that I am beginning to understand it but that it is still not easy to understand. I think a big part of this is that it is difficult for the reader to track what is happening at the batch level and what is happening as each item is being processed. As I understand it now, everything in the figure up to Explicit memory $\gamma^{t+1}$ is taking a batch and returning a small set of updated priors. These priors are then used to pull the representation of each image in the direction of the attractors represented by the priors. Crucially, If I understand correctly, all the attractors act on each image to adjust it in ways that enhance performance on the task. I think figure 1 could be drawn in a way that makes this clearer, perhaps with $\Xi^t$ at the left being processed through an upper track, with the Hopfield network receiving the lower track and emitting $\hat{\Xi}^t$  sending this forward. The caption should emphasize that the same small set of 32 upscaled attractors are used to adapt each image.**
> >
> > Yes, your understanding of the priors and the functioning of the attractors is correct. As suggested, we improved Figure 1.a for better clarity in batch-level processing. We added to the caption that the same small set of upscaled priors is utilized as attractors to adapt each image.
> >
> > **Q2: Also, I wondered if, at test time, it would make sense to freeze the explicit memory so that it is not being adjusted at all within a batch. Maybe that's what is already happening? In any case, this would dissociate the idea that these are priors with the idea that somehow they are being induced from the batch, and would make a useful ablation. I'd appreciate a comment on this before I finalize my score.**
> >
> > Thank you for pointing this out. During the test time, the explicit memory is frozen, functioning as fixed priors, and any memory update from the bottleneck attention will not be retained. We only compute $\gamma^{t+1}$ for the following pattern retrieval step in Hopfield networks for the current batch. To ensure a fair evaluation on the test dataset, the same explicit memory from the training time is utilized across test batches. We included a discussion on this point in Section 4.2 Bottleneck Attention with Limited Capacity to enhance clarity.
> >
> > We also conducted additional experiments on the case where these priors in the memory are frozen and also not utilized for the bottleneck attention during test time (no forward process to update the priors). We use only the Hopfield network to perform pattern retrieval based on the explicit memory, utilizing the current memory $\gamma^t$ without the forward process to obtain $\gamma^{t+1}$. We testified this ablation on the CIFAR, Triangle, and Pet datasets with AiT-Medium for three different seeds. Additionally, we conducted experiments on relational reasoning tasks (relational) with AiT-Small for three different seeds. The results suggest that relying solely on the Hopfield network retrieval without bottleneck attention to selectively extract patches from the test input leads to inferior performance. Utilizing the frozen memory as fixed priors for bottleneck attention during test time proves to be an effective strategy for enhancing performance while maintaining the integrity of the evaluation on the test data.
> >
> >
> > ### **Image Classification Tasks**
> > | Methods                            | **CIFAR10** | **CIFAR100** | **Triangle** | **Pet**   |
> > | ------------------------------------|---------|----------|----------|-------|
> > | AiT-Medium (frozen memory)          | 84.59   | 60.58    | 99.57    | 30.05 |
> > | AiT-Medium (without memory forward) | 84.50    | 60.56    | 99.57    | 28.68 |
> > &nbsp;
> >
> > ### **Relational Reasoning Task**
> > | Methods                              | **Relational** |
> > | --------------------------------------- | ---------- |
> > | AiT-Small (frozen memory)               | 76.82      |
> > | AiT-Small (without memory forward)      | 75.17      |

---

> > > ### Comment · Reviewer_H5y7 · 2023-11-22
> > > **Making sure I understand the new results**
> > >
> > > Thanks for these changes.  To make sure I understand, the 'without memory forward' is the procedure you used at test time in the standard AiT approach.  'frozen memory' corresponds to the new ablation and it performs as well as or slightly better than your standard testing approach.  You are not presenting additional results related to the sentence 'The results suggest that relying solely on the Hopfield network retrieval without bottleneck attention to selectively extract patches from the test input leads to inferior performance' -- this was already established in prior simulations.  Please include some mention of these new findings and let me know when the paper is updated.  I can't look further now, but will look again in about 2 hours, and will comment on the figure then.

---

> > > > ### Author Response · Authors · 2023-11-22
> > > > **The interpretation of the new results was added**
> > > >
> > > > Thank you very much for your suggestion. Please refer to Figure 8 in Appendix A.5 in P15 of the revised manuscript. Figure 8  shows (1) In AiT, although the explicit memory is frozen (depicted by filled dots), the memory forward process is enabled. However, the computed output of the multi-head cross-attention will not be used to update the memory, which is different from training time. (2) In the 'without memory forward' ablation, the entire memory forward process is disabled, leveraging only Hopfield networks for retrieval.
> > > >
> > > > The ablation result for the W/O bottleneck in Table 2 is the discussion on whether the bottleneck is used during **model training**. However, the discussion on the bottleneck here is regarding **test time** processing. Therefore, the sentence 'The results suggest that relying solely on the Hopfield network retrieval without bottleneck attention to selectively extract patches from the test input leads to inferior performance' refers to the new results of the 'without memory forward' ablation.
> > > >
> > > > Table 7 and Table 8 in P15 show that AiT performs as well as or better than the 'without memory forward' ablation.
> > > >
> > > > &nbsp;
> > > >
> > > > The modified parts of the manuscript include:
> > > >
> > > > P5: "During the test time, ... across all test batches."
> > > >
> > > > P8: "We demonstrate the without memory forward ... better than the without memory forward ablation."
> > > >
> > > > P15: Section "A.5 DISCUSSION ON THE TEST TIME SCHEME".

---

### Comment · Reviewer_H5y7 · 2023-11-21
**I'm pleased with the responses - I'll read the revised ms this evening**

I think I understand most of the responses to my questions and I am pleased with the further clarifications and new results.  I didn't realize that a revision had been uploaded but I see that the current version is different from the original and so I'll take a look later this evening.

---

### Meta-Review · Area_Chair_n7zK · 2023-12-16

**Metareview:**

This paper proposes a model called Associative Transformer (AiT), inspired by global workspace theory (GWT) and modern Hopfield networks (MHNs). They introduce a new global workspace layer which they stack after a Vision Transformer (ViT) module, showing some (modest) improvements in image classification tasks compared to ViT. Results on relational reasoning tasks seem less competitive. There was some disagreement among reviewers. The main weakness which have been pointed out by the reviewers is the modest grains brought by the proposed architecture which makes it unclear what the added complexity can buy us. The authors addressed some comments and improved the paper in the rebuttal period, and one of the reviewers (H5y7) seems to feel more positive about the paper now and wrote that they would change the score to "marginal above the threshold), although they haven't changed the score in the system. However, some of the concerns remain. Overall I feel this is a nice exploratory work which is promising but is not ready yet for publication in a top-tier conference. I encourage the authors to take the feedback from the reviewers in a future resubmission of their work.

**Justification For Why Not Higher Score:**

The modest grains brought by the proposed architecture and the fact that it requires the ViT component make it unclear what the benefits of the proposed GWL Overall I feel this is a nice exploratory work which is promising but is not ready yet for publication in a top-tier conference.

**Justification For Why Not Lower Score:**

N/A

---

### Decision · Program_Chairs · 2024-01-16

Reject